# Updating Insights into the Regulatory Mechanisms of Calcineurin-Activated Transcription Factor Crz1 in Pathogenic Fungi

**DOI:** 10.3390/jof8101082

**Published:** 2022-10-14

**Authors:** Yangyang Yang, Pengdong Xie, Yongcai Li, Yang Bi, Dov B. Prusky

**Affiliations:** 1College of Food Science and Engineering, Gansu Agricultural University, Lanzhou 730070, China; 2Department of Postharvest Science, Agricultural Research Organization, Volcani Center, Rishon LeZion 7505101, Israel

**Keywords:** calcium homeostasis, Crz1, fungi, cellular functions, molecular regulatory mechanisms, cross-talk

## Abstract

Ca^2+^, as a second messenger in cells, enables organisms to adapt to different environmental stresses by rapidly sensing and responding to external stimuli. In recent years, the Ca^2+^ mediated calcium signaling pathway has been studied systematically in various mammals and fungi, indicating that the pathway is conserved among organisms. The pathway consists mainly of complex Ca^2+^ channel proteins, calcium pumps, Ca^2+^ transporters and many related proteins. Crz1, a transcription factor downstream of the calcium signaling pathway, participates in regulating cell survival, ion homeostasis, infection structure development, cell wall integrity and virulence. This review briefly summarizes the Ca^2+^ mediated calcium signaling pathway and regulatory roles in plant pathogenic fungi. Based on discussing the structure and localization of transcription factor Crz1, we focus on the regulatory role of Crz1 on growth and development, stress response, pathogenicity of pathogenic fungi and its regulatory mechanisms. Furthermore, we explore the cross-talk between Crz1 and other signaling pathways. Combined with the important role and pathogenic mechanism of Crz1 in fungi, the new strategies in which Crz1 may be used as a target to explore disease control in practice are also discussed.

## 1. Introduction

Ca^2+^, as a second messenger, plays an important role in the regulation of biological function in cells. Unlike other second messengers, Ca^2+^ does not need to be synthesized but instead controls intracellular Ca^2+^ content through a series of complex regulatory mechanisms when responding to external signals. The CaN-Crz1 signaling cascade in fungal cells can be activated by different external stimuli, such as high temperature, low temperature, hypertonicity, alkalinity, oxidative stress, ethanol stress, light sources, antifungal drugs and others. The signal transduction mediated by Ca^2+^ can cause an instantaneous increase in intracellular Ca^2+^, which is generally considered to be the switch to turn on the signaling pathway [1,2,3,4,5,6,7]. The transient increase of intracellular Ca^2+^ content is caused by the entry of extracellular Ca^2+^ into cells through Ca^2+^ channel proteins Mid1 and Cch1 on the plasma membrane, or the release of Ca^2+^ from the intracellular calcium pool [8,9,10]. Intracellular free Ca^2+^ combines with calmodulin (CaM) to form a Ca^2+^/CaM complex and then activates calcineurin (CaN), which further dephosphorylates transcription factor Crz1 and allows it into the nucleus to regulate the expression of target genes [11]. The pathway is considered the Ca^2+^/calmodulin/Crz1 signaling pathway, also known as the CCS (calcium cell survival) pathway [8]. At present, the calcium signaling pathway has been systematically studied in mammals, parasites and yeasts [12,13,14,15,16,17,18]. Various components of the calcium signaling pathway play an important role in vascular development, axon outgrowth, stress response and glycogen synthesis in organisms [19,20,21,22,23,24]. This review briefly summarizes the calcium channels, calcium pumps and Ca^2+^ sensor proteins of the calcium pathway system in fungi, pointing out that the calcium homeostasis system is involved in a variety of life processes, such as cell growth, conidia production, stress response and maintenance of normal organelle function. We highlight recent findings on how transcription factor Crz1 regulates growth and development, stress responses, pathogenicity of pathogenic fungi and its regulatory mechanisms based on discussing the structure and localization of Crz1. In addition, cross-talk between Crz1 and other signaling pathways and how recent advances in our understanding of CaN-Crz1 signaling cascade might be used in practice to explore new strategies for disease control are also discussed.

## 2. Calcium Signaling Pathway in Fungal Cell

The calcium signaling system plays a very important regulatory role in the whole process of fungal growth and development. Imbalance in the calcium signaling system leads to abnormality of fungal cells in various aspects such as reproductive development, polar growth, cell differentiation and division, stress response and programmed death. Therefore, maintaining the stability of intracellular calcium levels is crucial for cell survival. Under normal physiological conditions, the concentration of cytoplasmic Ca^2+^ in fungal cells is in the low range of 50 to 100 nM [9,25]. The stability of Ca^2+^ levels in cells is controlled by a complex Ca^2+^ homeostasis regulatory system (Figure 1), which includes multiple Ca^2+^ channel proteins and pumps, as well as Ca^2+^ transporters, and many related proteins and enzymes in eukaryotes [9,25,26]. These components, mainly located on the plasma membrane or different subcellular organelles, are responsible for absorbing Ca^2+^ release from extracellular and intracellular calcium pools, thereby synergistically regulating the stability of Ca^2+^ levels in the cytoplasm and various organelles [27,28,29].

Two pathways have been reported to participate in extracellular Ca^2+^ uptake in fungi: the high-affinity Ca^2+^ transport system (HACS) and low-affinity Ca^2+^ transport system (LACS). The HACS, composed of Mid1 and Cch1, is responsible for Ca^2+^ uptake at low calcium concentrations (about 100 nM) [4,30,31]. Recently, Ecm7, a member of the PMP-22/EMP/MP20/Claudin superfamily of transmembrane proteins that includes γ-subunits of voltage-gated calcium channels, was identified as another subunit of HACS [32,33]. Cch1, the first Ca^2+^-related protein in the Ca^2+^/calmodulin/calcineurin/Crz1 signaling pathway, plays a critical role in regulating a variety of physiological activities activated by the calcium signaling system in fungal cells [34,35,36,37,38,39]. Mid1 and Cch1 are subject to feedback inhibition by calcineurin in a high calcium environment; then, the LACS plays a major role. The only known component of LACS to date is the membrane protein in Figure 1 [25,30,40]. The deletion of Figure 1 in fungi affects a wide range of cellular processes, such as sexual reproduction, mycelial growth, virulence and conidia production [41,42,43,44]. Recently, transient receptor potential (TRP) channels were found among mammals, flies, worms, ciliates, Chlamydomonas and yeasts [45]. The TRP channels act as sensors for various stresses, including temperature, pH, osmolarity and nutrient availability [46,47,48,49]. The first calcium-permeable TRP, initially isolated from *Arabidopsis thaliana*, can be activated by hyperosmotic shock and, therefore, was named calcium-permeable stress-gated cation channel 1 (CSC1) [47], which includes the PenV protein of *P. chrysogenum* and CefP of *A. chrysogenum* [50]. The Yvc1 channel protein located on the tonoplast is a homologue of mammalian transient receptor potential (TRP) channel protein responsible for the release of Ca^2+^ from the vacuole into the cytoplasm [10,51,52]. FLC was recently proposed as a member of the FLC family required for importing FAD into the endoplasmic reticulum, and it represent a conserved fungal gene family of integral membrane protein, spanning a TRP-like domain [49,53]. Some studies suggest FLC could act as either a calcium sensor or directly as a calcium channel [49].

There are many kinds of calcium pools in fungal cells, such as endoplasmic reticulum, Golgi apparatus and vacuoles. Different calcium pumps are distributed in these calcium pools, and are responsible for transporting Ca^2+^ from the cytoplasm to various organelles against the concentration gradient. For fungal cells, vacuoles rather than endoplasmic reticulum are the most important calcium pools, where the concentration of Ca^2+^ is about 10^4^ times that of cytoplasmic [54,55]. This large amount of Ca^2+^ storage is maintained by the action of two transporter proteins, Ca^2+^-ATPase Pmc1 and Ca^2+^/H^+^ exchanger Vcx1 [10,26,56,57,58,59,60]. Vcx1 belongs to the CAX superfamily of calcium-permeable ion exchangers [61,62,63]. When there is a burst in the cytoplasmic content of calcium, the Vcx1 transporter sequesters the calcium into the vacuoles. In addition to calcium, the Vcx1 protein transports Mn^2+^ ions, thus allowing *S. cerevisiae* to grow in high concentrations of either calcium or manganese ions [64]. Pmr1 (Plasma membrane ATPase related) is the first member of the secretory pathway Ca^2+^-ATPase (SPCA) family, which mediates the transport of Ca^2+^ and Mn^2+^ in Golgi under normal physiological conditions [56,65,66,67,68,69].

In order to precisely regulate intracellular calcium signals, organisms have also evolved several calcium-sensing proteins to respond to different ranges of Ca^2+^ concentration levels [70]. CaM, located downstream of phospholipase C [71] in the calcium signaling pathway, is a very important Ca^2+^ sensor that can sense the change of intracellular Ca^2+^ concentration and regulate a series of downstream target proteins by binding with Ca^2+^ [72,73,74]. CaN, as a Ca^2+^ and CaM dependent serine/threonine protein phosphatase, is composed of the catalytic subunit CNA and the regulatory subunit CNB [75,76,77,78,79], and is the central mediator of the Ca^2+^/calmodulin/calcineurin/Crz1 signaling pathway. In fact, calcineurin regulates the activity of diverse calcium transporters on the plasma membrane and is mainly responsible for calcium homeostasis [80]. Upon Ca^2+^ presence, the activated CaM binds to the CNA and CNB complexes to form a fully activated trimer [81], and then activated CaN dephosphorylates Crz1 and transfers it to the nucleus to regulate the expression of Crz1-dependent genes [11,82]. In fungi, the CaN-Crz1 signaling pathway is also conserved and involved in many biological processes, such as cell growth, infection structure differentiation, cell wall integrity, pathogenicity and stress response [83,84,85,86,87] (Figure 1). The Figure 1 only depicts genes in the calcium signaling pathway that are closely linked to Crz1 or directly regulated by Crz1.

## 3. Calcineurin-Activated Transcription Factor Crz1

### 3.1. Structure and Localization of Calcineurin Responsive Transcription Factor Crz1

Crz1 is the earliest identified downstream target protein of CaN in the Ca^2+^/CaM-CaN signaling cascade reaction [88]. Crz1 contains six important domains, including the C2H2 zinc finger DNA-binding domain, the calcineurin-docking domain (CDD), the serine-rich region (SRR), the nuclear export signal (NES), nuclear localization signal (NLS), and calcineurin docking motif (docking site to calcineurin, PIISIQ) [11,89,90,91,92,93,94]. The C2H2 zinc finger domain can bind to some target gene promoter response elements, which are called CDREs (calcineurin-dependent response elements). Therefore, a gene promoter sequence with this element is likely to be regulated by Crz1 [88]. The docking motif PIISIQ reported in *Saccharomyces cerevisiae* is the site of interaction between CaN and Crz1 [92]. The SRR structural domain, a serine-rich region containing several serine residues, is the target site for dephosphorylation of Crz1 by calcineurin and determines the localization and phosphorylation level of Crz1 [82,95]. Without external stresses or stimulus, Crz1 is localized in the cytoplasm, while upon increased Ca^2+^ concentration, CaN is activated to dephosphorylate Crz1, and then dephosphorylated Crz1 relocates to the nucleus for regulating targeted genes. This localization can be reversed by inhibitors, such as cyclosporine A, which inhibits CaN activity and redistributes Crz1 to the cytoplasm [11]. In addition, Crz1 can be phosphorylated in the presence of protein phosphokinase. The homologous protein Hrr25 of casein kinase 1 in mammals was detected by the high-throughput protein chip method [96]. In *S. cerevisiae*, Hrr25 plays a role in responding to DNA damage, mitosis and vacuole transport. In vivo, Hrr25 can bind to Crz1 and phosphorylate it to change its localization. The phosphorylated Crz1 is transported to the cytoplasm to avoid its accumulation in the nucleus. The ability of Crz1 to transport between cytoplasm and nucleus is regulated by NLS and NES. NLS and NES are able to form complexes with cellular input or output proteins, respectively. There is an NLS at the C-terminus of dephosphorylated Crz1, which can bind to the nuclear input protein Nmd5. Therefore, Nmd5 is responsible for transporting Crz1 to the nucleus. Interaction between NES and nuclear export protein Msn5 is responsible for nuclear export of phosphorylated Crz1 [73,82,97]. However, different Crz1 nuclear input and output proteins have recently been found in the industrial fungus *Penicillium oxalicum* [98]. Using tandem affinity purification combined with mass spectrometry (TAP-MS), no *Msn5* homologue was found in *P. oxalicum* instead of the nuclear transporter Los1. Therefore, it is more likely that PoCrz1 is exported from the nucleus through Los1 than through Msn5. Los1 and Msn5 play some overlapping roles in nuclear output [99]. In addition, *PoCrz1* enters the nucleus through Srp1 rather than Nmd5 [98]. These findings suggest that Crz1 transportation between cytoplasm and nucleus is also finely regulated.

### 3.2. Transcription Factor Crz1 Regulates Fungal Growth and Development

The transcription factor Crz1 regulates target genes and proteins through the calcium signaling cascade pathway and ultimately affects fungal growth, development and pathogenicity. Deletion of *Crz1* resulted in abnormal development of vegetative growth of most pathogenic fungi. The *ΔBc**Crz1* mutant in *Botrytis cinerea* demonstrated impaired mycelial growth and abnormal branching on CM medium [90]. Similarly, the vegetative growth of *ΔFg**Crz1* in *Fusarium graminearum* and *ΔAn**Crz1* in *Aspergillus nidulans* shows severe defects [100,101]. However, in *Penicillium digitatum*, *Aspergillus fumigatus*, *Magnaporthe grisea* and *Verticillium dahliae*, the absence of *Crz1* has no significant effect on their vegetative growth [91,102,103,104]. In human pathogenic fungus *Candida lusitaniae*, the deletion of *Crz1* is associated with the loss of the ability to transform from yeast to hyphal morphology [105]. The cell structure of the WT and *Crz1* mutants in *Candida glabrata* was observed via transmission electron microscope and it was found that compared with WT, the *Crz1* mutants demonstrated irregular plasma membrane structure and abnormal organelles [106]. Formation and development of fungal conidia require Crz1. For example, after knocking out *Crz1* in *B. cinerea*, the *ΔBc**Crz1* cannot produce sporophores or conidia [90]. The *ΔFg**Crz1* in *F. graminearum* was unable to form perithecium, which affected its sexual development [100]. After the deletion of *Crz1*, *Valsa pyri* could not form a fruiting body structure [107]. Other studies have reported that the *A. nidulans* could open the calcium channel through the pressure sensor on the cell wall, and the CNA/Crz1 complex was activated, thereby promoting the polar growth of mycelia [108]. In a word, Crz1 is involved in various physiological functions of fungi, which we summarized in Table 1.

### 3.3. Transcription Factor Crz1 Is Essential for Fungal Pathogenicity

The virulence regulated by Crz1 was first studied in *C. albicans* [94,109], a human pathogenic fungus, and it was confirmed that the deletion of *Crz1* would reduce the virulence. Crz1 is also associated with the virulence of other *Candida* species. In emerging fungal pathogens *C. lusitaniae* and *C. glabrata*, it has been shown that the signal transduction pathway of CaN-Crz1 can control the virulence of the systemic infection model in mice [105,106,110]. Interestingly, the effect of Crz1 on virulence was also related to the specific niche of the host. For example, Crz1 is particularly important for murine eye infection, but it does not play a role in the murine urinary tract infection model [106]. It is well known that Crz1 is also necessary for mycelial growth, morphological transformation and spore and appressorium formation of filamentous fungi [89,90,91,102,103,104], on top of being the precursor for the formation and maintenance of pathogenicity of pathogenic fungi. In *Magnaporthe oryzae*, compared with the WT, the reduced pathogenicity of the *Crz1* knockout strain is mainly due to the decreased swelling pressure of appressorium, which leads to osmotic damage [89]. The reduction of appressorium swelling found in *ΔMg**Crz1* is reported to be caused by disruption of lipid metabolism [103]. In *B. cinerea*, the absence of *Crz1* can cause defects in cell wall and membrane integrity, thus weakening the ability of hyphae to penetrate plant tissues [90]. The significantly decreased pathogenicity of the *ΔFg**Crz1* in *F. graminearum* was suggested to be associated with impaired toxin DON biosynthesis [100]. In summary, through these studies on the pathogenic infection mechanisms of fungi pathogens, it was found that although Crz1 played a conservative role in fungi virulence or pathogenicity, the pathogenic mechanisms were different.

### 3.4. Transcription Factor Crz1 Involved in Fungal Stresses Responses

Fungi are frequently exposed to a variety of environmental stresses, including metal ions, oxidative stress, pH and cell wall interference agents. In order to cope with these environmental stresses, fungi evolve various strategies to quickly sense these signals, and then reduce the damage caused by environmental stresses. The transcription factor Crz1 is activated by stress-induced elevated Ca^2+^ levels and regulates the expression of related genes in response to these stresses. Crz1 is involved in the response of fungi to various stresses, as shown in Table 2.

#### 3.4.1. Transcription Factor Crz1 in Ion Stress Response

In fungi, the *Crz1* mutant is sensitive to ion stress, especially hypersensitivity to Ca^2+^, which has been reported in several studies [89,90,91,102,103,111,112] and may be due to the dephosphorylated Crz1 being transferred into the nucleus to induce the expression of multiple genes related to calcium ion stress, such as PMC and PMR [69,93,113,114]. However, sensitivity to other cation ions stresses such as Na^+^, Li^+^, Mg^2+^ and Mn^2+^ varies among *Crz1* deleted fungal species. In *A. fumigatus*, the *ΔAf**Crz1* demonstrated strong sensitivity to Mn^2+^, but low sensitivity to Na^+^ and Li^+^ [91]. For *M. grisea*, the *ΔMg**Crz1* was insensitive to Na^+^, Li^+^ and Mn^2+^ [89,103]. On the contrary, the *ΔBc**Crz1* mutant demonstrated a strong sensitivity to these four ion stresses. In addition, it was found that the addition of Mg^2+^ restored growth defects and cell wall integrity in the *ΔBc**Crz1* of *B. cinerea* [90]. These data suggest that ion stress responses and ion homeostasis regulated by Crz1 are a common feature in fungi, although there was species specificity.

#### 3.4.2. Transcription Factor Crz1 in Oxidative Stress Response

Yeast glutathione peroxidase GPX2 is a part of the antioxidant system that protects cells from oxidative stress. The expression of GPX2 induced by H_2_O_2_ is strictly regulated by transcription factor YAP1 and response regulator SKN7 [115,116]. Meanwhile, SKN7 has been found to be a multicopy enhancer of CaN-Crz1 dependent transcription in yeast, and SKN7 regulates calcineurin signaling by stabilizing Crz1 through direct protein–protein interaction [117]. The sensitivity of Crz1 to oxidative stress was also confirmed in *B. cinerea* [90], *M. acridum* [118] and *P. digitatum* [102]. The specific regulatory role of Crz1 in fungal pathogen response to oxidative stress needs to be further elucidated.

#### 3.4.3. Transcription Factor Crz1 in pH Stress Response

Crz1 is essential for tolerance to high pH conditions in yeast. Upon stimulation of alkaline conditions, Ca^2+^ enters the cytoplasm through the Cch1-Mid1 channel and then activates CaN to dephosphorylate Crz1 into the nucleus to induce several alkaline pH-responsive gene expressions, including *ENA1*, *PHO84*, *PHO89* and *PHO12* [119,120]. The colony growth rate of *ΔBc**Crz1* slowed down under extreme pH (3 or 9). Interestingly, exogenous Mg^2+^ addition could restore the growth phenotype at pH 9, but the *ΔBcCrz1* growth defect phenotype did not recover at pH 3 [90].

#### 3.4.4. Transcription Factor Crz1 in Cell Wall Interference Agents

The growth of *Crz1* mutants in *P. digitatum*, *M. oryzae* and *B. cinerea* were seriously damaged in the medium containing cell wall inhibitors [89,90,102]. However, compared with the WT, the mycelial growth of *ΔVp**Crz1* was significantly increased on CM agar medium containing SDS, CR or CFW, which was inconsistent with previous reports. It was suggested that *Vp**Crz1* acted as a negative regulator of cell wall stress in *V. pyri* [107]. Similarly, the *Crz1* mutant demonstrated resistance to SDS in human pathogenic fungus *Candida lusitaniae*, indicating that Crz1 negatively regulated cell membrane integrity, while Crz1 was found to respond to SDS by an unknown mechanism independent of CaN [105]. 

In addition, the involvement of Crz1 in fungal stress resistance was also reflected in the tolerance of antifungal drugs, temperature and ethanol. It has been reported that the damage of Crz1 in *S. cerevisiae* increases its sensitivity to azole drugs, while its overexpression reduces the sensitivity [7]. Similarly, Crz1 is responsible for azole resistance in *P. digitorum* as well as *ΔPd**Crz1* reduced imidazole and difenoconazole tolerance [102]. In *C. neoformans*, Crz1 homologous phospholipid binding protein Cts1 was identified as a CaN substrate for high-temperature stress [121]. The *ΔCg**Crz1* in *C. glabrata* could not grow as normally as the WT at 40 °C [106]. Ethanol was a common stress source in yeast. The cells lacking *Crz1* demonstrated poor adaptation to ethanol stress, while the multi-copy plasmid of Crz1 improved the tolerance to ethanol stress. Therefore, Crz1 was crucial for the survival of yeast cells under ethanol-induced stress [122]. It has been demonstrated in *C. neoformans* that Crz1 is involved in cell survival, biofilm formation and fluconazole sensitivity in the hypoxic environment [123].

**Table 1 jof-08-01082-t001:** Regulatory roles of transcription factor Crz1 in fungi.

Fungal Species	Cellular Functions of Crz1	Selected References
*Alternaria alternata*	Infection structure differentiation Pathogenicity Vegetative growth Stress tolerance Cell wall integrity Melanin production Calcium homeostasis	[86]
*Magnaporthe oryzae*	Conidiation Ionic homeostasis Cell wall integrity Virulence	[89]
*Botrytis cinerea*	Vegetative growth Mycelial morphology Conidiation Cell wall integrity Virulence	[90]
*Fusarium graminearum*	Vegetative growth Sexual development Toxin synthesis Stress responses Virulence	[100]
*Penicillium digitatum*	Conidiation Virulence DMI resistance	[102]
*Magnaporthe grisea*	Conidiation Appressorium formation Calcium tolerance Melanin production Lipid metabolism Virulence	[103]
*Verticillium dahliae*	Microsclerotia development Melanin accumulation Cell wall integrity Virulence	[104]
*Candida lusitaniae*	Cell wall integrity ER stress Pseudohyphal growth Ca^2+^ homeostasis Virulence	[105]
*Candida glabrata*	Thermotolerance cell morphology Virulence ER stress tolerance	[106]
*Valsa pyri*	Fruiting body formation Mycelial morphology Virulence Cell wall perturbing agents resistance	[107]
*Cryptococcus neoformans*	Hypoxic adaptation Inbiofilm formation Cell wall integrity Fluconazole tolerance	[123]

**Table 2 jof-08-01082-t002:** Stress responses regulated by transcription factor Crz1 in fungi.

Environmental Stresses	Fungal Species	Selected References
Ion stress	*Magnaporthe oryzae*	[89]
	*Botrytis cinerea*	[90]
	*Aspergillus fumigatus*	[91]
	*Penicillium digitatum*	[102]
	*Magnaporthe grisea*	[103]
	*Torulaspora delbrueckii*	[111]
	*Aspergillus nidulans*	[112]
Oxidative stress	*Botrytis cinerea*	[90]
	*Penicillium digitatum*	[102]
	*Saccharomyces cerevisiae*	[115,116]
	*Metarhizium acridum*	[118]
Alkaline stress	*Botrytis cinerea*	[90]
	*Saccharomyces cerevisiae*	[119,120]
Cell-wall-perturbing agents	*Magnaporthe oryzae*	[89]
	*Botrytis cinerea*	[101]
	*Penicillium digitatum*	[102]
	*Candida lusitaniae*	[105]
Antifungal agents	*Saccharomyces cerevisiae*	[7]
	*Penicillium digitatum*	[102]
High temperature stress	*Candida glabrata*	[106]
	*Cryptococcus neoformans*	[121]
Ethanol stress	*Saccharomyces cerevisiae*	[122]
Hypoxic stress	*Cryptococcus neoformans*	[123]

### 3.5. Molecular Regulatory Mechanisms of Transcription Factor Crz1 in Pathogenic Fungi

The zinc finger domain of Crz1 specifically binds to the 24 bp CDREs sequence to initiate target gene expression [88,124]. In *S. cerevisiae*, the core consensus site for Crz1 binding is 5′-GNGGCKCA-3′ [93], and the putative DNA common sequence bound by Crz1 in *Trichoderma reesei* was identified as 5′-GDGGCKBNB-3′ [125]. Therefore, we hypothesize that 5′-GNGGCK-3′ is a common sequence of Crz1-binding DNA. The target genes involved in ion homeostasis, cell wall maintenance, lipid synthesis, protein degradation and glucose metabolism are regulated by Crz1. Several studies have identified species-specific genes regulated by Crz1, and Crz1 can also be used as an inducer or inhibitor of gene expression. Crz1 is necessary for PMC and PMR to respond to Ca^2+^. PMC and PMR belong to the P-type ATPase superfamily, which can obtain energy by hydrolyzing ATP to drive Ca^2+^ transport from the cytoplasm to the vacuole and the Golgi, respectively, to maintain intracellular calcium homeostasis [67,113]. In fungi, the expression of *PMC* and *PMR* genes is significantly induced in response to Ca^2+^, but the expression levels are not highly activated in the *Crz1* mutants [89,91,102]. The reduced expression of these ATPases prevented the normal translocation of excess Ca^2+^ from the cytoplasm to various organelles, resulting in a disruption of calcium homeostasis, which may account for the sensitivity of *Crz1* mutants to Ca^2+^. *ENA1*, *ENA2*, and *ENA3* belong to the encoding plasma membrane Na^+^/Li^+^-ATPase, which are necessary for yeast survival under high Na^+^ and Li^+^ concentrations, and their expression is also induced by CaN in a Crz1-dependent manner [93,126]. In addition, other genes involved in ion homeostases such as *MEP1*, *ENB1*, *PHO84*, *PHO89* and *KHA1* are also regulated by CaN-Crz1 pathway [93]. Under external stress stimulation, the *β*-1,3 glucan synthase (FKS) and the chitin synthase (CHS) are essential for maintaining cell wall integrity. In the *Crz1* mutant, both *FKS* and *CHS* expression are disrupted [88,90,92,112]. Other genes involved in maintaining cell wall integrity such as *CRH1*, *RHO1*, *SCW10* and *KRE6* are also regulated by the CaN-Crz1 pathway [93]. In *P. oxalicum*, an industrial fungus, Crz1 plays a role in cellulase synthesis by regulating the expression of cellulose decomposition genes such as *cbh1*, *eg1* and *eg2* [98]. Expression of genes related to lipid and sterol metabolism such as *SUR1*, *CSG2*, *YSR3*, *ERG26*, *HES1* and *PLB3*, as well as genes involved in vesicular transport such as *GYP7*, *YPT53*, *YIP3*, *PEP12*, *RVS161*, *SHE4*, *CVT17*, *CVT19* and *VPS36*, all of which are regulated by Crz1, thus enables cells to maintain normal membrane function and complete the process of substance delivery to the cell surface [93]. However, studies have found that not all Crz1 functions depend on CaN. As demonstrated in *C. neoformans*, Crz1 exhibits a specific CaN-independent response to different environmental stress stimuli [127,128], Furthermore, in *C. dubliniensis*, Crz1 regulates haptotropic (surface-sensing) responses independently of CaN [129].

### 3.6. Cross-Talk between Transcription Factor Crz1 and Other Signaling Pathways

At present, it has been found that Crz1, a downstream transcription factor of the calcium signaling pathway, is not only related to calcium signaling but also participates in the transcriptional regulation of other signaling pathways. The cell wall integrity (CWI) pathway, one of the MAPK cascades pathways, maintains cell wall integrity by mediating cell wall biosynthesis. Since cell wall integrity is critical for cells to cope with environmental stress, CWI pathways need to cross-talk with other proteins or pathways to enhance their transduction ability [130,131]. Numerous studies have found that Crz1 maintains cell wall integrity by regulating genes involved in *CHS* and *FKS* biosynthesis [88,90,92,112,132,133]. Therefore, it is inferred that Crz1 cooperates with the CWI pathway to regulate cell wall integrity.

The high-osmolarity glycerol (HOG) pathway is used to regulate various stress genes for osmotic protection, and activation of this pathway is regulated by two upstream branches, one mediated by the *Sho1* sensor and the other by a system consisting of *Sln1*, *Ypd1* and *Ssk1* [134,135,136,137,138]. At the same time, Crz1 participates in the regulation of ion osmotic homeostasis by mediating the expression of ion transport genes [89,91,92,93,102,116]. Shitamukai et al. [139] found that there was a crosstalk relationship between the HOG and the CaN-Crz1 signaling pathway, and proved that there was an antagonistic effect between them. The CaN-Crz1 signaling pathway is involved in the downregulation of the HOG pathway by regulating the *Sln1* branch. In addition, the cyclic adenosine monophosphate-protein kinase A (cAMP-PKA) pathway is also antagonistic to the CaN-Crz1 signaling pathway. It was found that Crz1 is a substrate for PKA, which is functionally opposite to the CaN signaling pathway, and PKA can directly phosphorylate Crz1 to inhibit its nuclear localization and activity [140].

In *S. cerevisiae*, *Neurospora crassa* and mammals, it has been shown that external signals are sensed by G protein-coupled receptors (GPCRs) [141,142]. After sensing the stimulation of external signals, membrane binding receptors trigger G protein to dissociate Gα subunit from Gβ/γ subunit. The released Gα subunit activates phospholipase C (PLC), which hydrolyzes inositol-4,5-diphosphate (PIP2) to generate two important messenger molecules, diacylglycerol (DAG) and inositol-1,4,5-triphosphate (IP3) [143]. Among them, IP3 can stimulate endoplasmic reticulum, vacuoles, Golgi and other organelles to release Ca^2+^, thereby activating calcium signaling pathway [144,145,146]. Therefore, we propose a correlation between CaN-Crz1 signaling and the G protein-coupled receptor system (Figure 1). It was reported that glucose addition stimulates a rapid increase in free calcium level in yeast, thus activating the calcium signaling pathway [147,148]. Furthermore, Plc1p is essential for glucose-induced calcium increase. Studies suggest that Plc1p is activated by glucose firstly, and then lead to cleavaging PIP2 and generating IP3 for raising the calcium level in the cytosol [148]. However, in strains with a deletion in the *GPR1* or *GPA2* genes, the calcium influx induced by addition of high glucose was inhibited, which suggests the physiological process requires the Gpr1p/ Gpa2p receptor/G protein-coupled (GPCR) complex [149,150].

In *S. cerevisiae*, DNA microarray data indicated that a total of 150 genes responded to the alkaline pH environment, but the expression of many alkali-induced genes was inhibited in the CaN or Crz1 mutants, suggesting that calcium signaling is involved in the alkaline stress response [120]. The Rim101 signal transduction pathway is responsible for the adaptation of *C. albicans* to the alkaline environment [151]. Wang et al. [2] confirmed that *C. albicans* activated the calcium influx system in response to alkaline stress, and both Rim101 and Crz1 were involved in the activation of PHO89 promoter induced by alkaline stress, indicating that Rim101 and Crz1 signaling pathways had potential chelating effects in *C. albicans* response to alkaline stress. In addition, the interaction between CaN-Crz1 and heat shock proteins (Haps) is involved in response to different environmental stress conditions [152]. Hsp90 physically interacts with calcineurin and mediates echinocandin resistance in *C. albicans* [153]. In *A. fumigatus*, the MAPK, Hsp90, and calcineurin signaling pathways are linked and play a role in drug resistance and development [154]. These data show that cross talk between calcineurin-Crz1 and other signaling pathways is common but the detailed molecular mechanisms need to be investigated further.

## 4. Conclusions and Prosect

In response to complex environmental stimuli, fungi regulate multiple cellular metabolic processes by sensing intracellular Ca^2+^ concentration changes and then activating expressions of target genes. As an important transcription factor downstream of the calcium signaling pathway, Crz1 is highly conserved in fungi and plays a critical role in growth, development, tolerance to stress conditions and pathogenicity. Although our insight into Crz1 biological function has recently advanced with unprecedented speed, there are still some open research problems that urgently need to be addressed: (1) the specific molecular mechanism of Crz1 in transcriptional regulation of target genes in calcium homeostasis system still needs to be further elucidated, (2) It is necessary to further carry out genetic and biochemical analysis experiments combined with transcriptome sequencing technology to understand the metabolic pathway regulated by the transcription factor Crz1 in fungi, (3) new, environmentally safe, species-specific strategies for disease control, such as RNA interference (RNAi) technology, should be explored based on clarifying its regulatory mechanism of Crz1.

## Figures and Tables

**Figure 1 jof-08-01082-f001:**
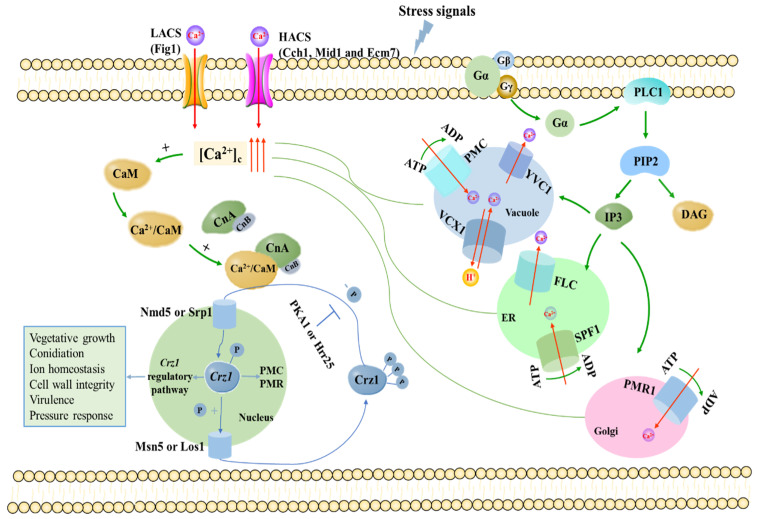
Pattern of calcium homeostasis system in fungi. When the cytosolic Ca^2+^ concentration increases, calmodulin activates calcineurin, which in turn dephosphorylates Crz1. Crz1 is then imported into the nucleus and induces or represses expression of its target genes. HACS: high-affinity calcium system; LACS: low-affinity calcium system; [Ca^2+^]_c_: cytosolic calcium concentration; CaM: calmodulin; CnA: calcineurin catalytic subunit; CnB: calcineurin regulatory subunit; Crz1: calcineurin responsive transcription; PMC: plasma membrane Ca^2+^-ATPase; PMR: plasma membrane ATPase-related pump; ER: endoplasmic reticulum; FLC: flavin carriers; PLC1: phospholipase C; PIP2: inositol-4,5-diphosphate; IP3: inositol triphosphate; DAG: diacylglycerol.

## Data Availability

Not applicable.

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
