# Peer review of "Updating Insights into the Regulatory Mechanisms of Calcineurin-Activated Transcription Factor Crz1 in Pathogenic Fungi"

_jof, 2022, doi:10.3390/jof8101082_

Round 1

Reviewer 1 Report

The manuscript 'Calcium homeostasis and the regulatory mechanism of calcineurin activated transcription factor Crz1 in pathogenic fungi: A review' by Yangyang Yang , Pengdong Xie , Yongcai Li, Yang Bi , Dov Prusky aims to be a comprehensive review of the field regarding calcineurin and Crz1 role in fungal pathogenicity. Actually, there is a very limited description of calcium homeostasis and also of the role of calcineurin in pathogenic fungi, while the review focuses more on Crz1 regulation in pathogenic fungi. The title should be more focused on the real topic of the review if the authors' intention is to focus on this topic specifically.

Major remarks.

- Chapter 2. The description of calcium transporters in fungi is limited to the classical three systems that have been well studied and known for decades. The review should mention the last findings on this topic, since several reviews already fully described these systems but novel characteristic were recently enlightened for Yvc1 also in pathogenic fungi. Moreover, novel families of calcium channels have been proposed more recently in pathogenic fungi as well as in model yeast systems. For example, there is no mention of other TRP family transporters present in fungi (for example the polycystin lyke FLC/spray family described in https://doi.org/10.3934/molsci.2016.4.505 , which in S. cerevisiae is directly involved in Crz1 regulation as reported in https://doi.org/10.1016/j.ceca.2014.12.003, or other that were reported to be involved in pathogenesis such as in https://doi.org/10.1186/s12866-018-1193-9 and in https://doi.org/10.3390/jof7110920 but also https://doi.org/10.1091/mbc.e18-04-0270 ) or the CSC group of proteins that have been fully characterized as calcium-permeable stress-gated ion channel proteins (https://doi.org/10.3390/genes13081450) also related to pathogenicity, or MCU that, although absent in budding yeast, is actually present in several fungi such as N. crassa or Aspergillus spp., nor of the CAX family members that have been known in fungi for ten years now. The authors should not attempt to describe calcium homeostasis if they are not willing to describe it fully and they should declare that they are only mentioning a part of the actors and which criteria they are using to define which systems they choose to describe. Most of all, do not mention in the title that your review is about homeostasis if you do not intend to fully describe it.

- Line 314. 'Therefore, we propose a correlation between 314 CaN-Crz1 signaling and the G protein-coupled receptor system (Fig. 1)'. Actually, such a correlation was previously proposed in S. cerevisiae, where the activation of a calcium pulse upon glucose re-addition was proved to be dependent on both Gpr1/Gpa2 signalling and Plc1 activity (references are completely lacking in the review on this pathway, such as the following doi: https://doi.org/10.1016/j.ceca.2011.03.006 ; https://doi.org/10.1007/s00294-003-0465-5 ; https://doi.org/10.1016/s0014-5793(02)02806-5 ; https://doi.org/10.1016/s0167-4889(98)00099-8 .) The authors should at least mention that this correlation is strengthened by previous results.

The reviewer has no minor remarks for the authors to address, the manuscript is well organized and correctly written.

Reviewer 2 Report

The paper has overall good quality. I have few comments:

more important:

1) The figure is difficult to understand as part of it is too small and is difficult to read anything...

2) Oly few references comes from last few years...Lack for example information about connection with heat shock proteins  during stress regulation...Roy, A., Tamuli, R. Heat shock proteins and the calcineurin-crz1 signaling regulate stress responses in fungi. Arch Microbiol 204, 240 (2022). https://doi.org/10.1007

So references and paper should be actualized according to newly published papers...

small comment

- instead of the  carbon terminus better to use the C-terminus or the  carboxy-terminus

Round 2

Reviewer 1 Report

The revised version of the manuscript has a different title, being 'Updating insights into the regulatory mechanisms of calcineurin-activated transcription factor Crz1 in pathogenic fungi.

The revised version now thoroughly, albeit briefly, summarizes the state of the art of calcium homeostasis in fungi but I appreciate that the title is better focused on the main topic of the manuscript. The paper is well written and organized and is quite an enjoyable update on the field.

I only have some minor remarks about imprecision in the implemented information.

- Line 91-94. 'studies have shown that this calcium release likely originates from the secretory compartments', the authors seem to implicate that FLC gene products are involved in hyperosmotic shock response, whereas they are actually engaged in hypotonic shock-induced calcium release.

- Crz1 in the title should not be in italics since it indicates the protein, I guess? The same happens throughout the whole manuscript
